# Malignant Pleural Effusion and Its Current Management: A Review

**DOI:** 10.3390/medicina55080490

**Published:** 2019-08-15

**Authors:** Kristijan Skok, Gaja Hladnik, Anja Grm, Anton Crnjac

**Affiliations:** 1Faculty of Medicine, University of Maribor, Institute of Biomedical Sciences, Taborska Ulica 8, SI-2000 Maribor, Slovenia; 2Faculty of Medicine, University of Maribor, Taborska Ulica 8, SI-2000 Maribor, Slovenia; 3Department of thoracic surgery, University Medical Centre Maribor, Ljubljanska 5, SI-2000 Maribor, Slovenia

**Keywords:** thoracic surgery, malignant pleural effusion, lung cancer, breast cancer, pleural carcinosis, treatment guidelines, LENT score

## Abstract

Malignant pleural effusion (MPE) is an exudative effusion with malignant cells. MPE is a common symptom and accompanying manifestation of metastatic disease. It affects up to 15% of all patients with cancer and is the most common in lung, breast cancer, lymphoma, gynecological malignancies and malignant mesothelioma. In the last year, many studies were performed focusing on the pathophysiological mechanisms of MPE. With the advancement in molecular techniques, the importance of tumor-host cell interactions is becoming more apparent. Additionally, the process of pathogenesis is greatly affected by activating mutations of *EGFR*, *KRAS*, *PIK3CA*, *BRAF*, *MET*, *EML4/ALK* and *RET*, which correlate with an increased incidence of MPE. Considering all these changes, the authors aim to present a literature review of the newest findings, review of the guidelines and pathophysiological novelties in this field. Review of the just recently, after seven years published, practice guidelines, as well as analysis of more than 70 articles from the Pubmed, Medline databases that were almost exclusively published in indexed journals in the last few years, have relevance and contribute to the better understanding of the presented topic. MPE still presents a severe medical condition in patients with advanced malignancy. Recent findings in the field of pathophysiological mechanisms of MPE emphasize the role of molecular factors and mutations in the dynamics of the disease and its prognosis. Treatment guidelines offer a patient-centric approach with the use of new scoring systems, an out of hospital approach and ultrasound. The current guidelines address multiple areas of interest bring novelties in the form of validated prediction tools and can, based on evidence, improve patient outcomes. However, the role of biomarkers in a clinical setting, possible new treatment modalities and certain specific situations still present a challenge for new research.

## 1. Introduction

Malignant pleural effusion (MPE) is an effusion, characterized by the presence of malignant cells [1]. MPE is a common manifestation in patients with metastatic disease and can occur in 15% of patients with cancer [2,3]. It is most common in lung cancer (LC), followed by breast cancer (BC), lymphoma, gynecological cancers and malignant mesothelioma [4]. There are 150,000 new cases of MPE yearly in the United States and 100,000 in Europe [4]. Patients have an overall survival (OS) rate of 3–12 months after the initial diagnosis of MPE [2]. Frequent symptoms in most patients with MPE include dyspnea, cough and chest pain [2]. The level of severity of the accompanying symptoms depends on the amount/volume of the effusion and the patients’ cardiopulmonary state. Just recently, after seven years, the new guidelines for the management of MPE were published by the European respiratory society (ERS/EACT) and the collaboration of American Thoracic Society, Society of Thoracic Surgeons, and Society of Thoracic Radiology (ATS/STS/STR) [5]. At the same time, new findings regarding the pathophysiology of MPE were discovered. This review offers an overview of the newest guidelines and findings regarding MPE management, as well as pathophysiological background. 

## 2. Pleural effusion

### 2.1. Anatomy and Physiology of Pleural Effusion

The pleural space forms between the fourth and seventh week of embryonal development. The visceral and parietal pleurae both derive from the lateral plate mesoderm which splits the somatopleuric mesoderm into two layers forming the parietal membrane and the splanchnopleuric mesoderm of the visceral membrane. Both are of vital importance for the homeostasis of the pleural space [6]. The pleural mesothelium, derived from the embryonic mesoderm, is a monolayer of mesothelial cells that blanket the chest wall and lungs on the parietal and visceral surfaces, respectively. These pavement-like cells are similar in cytologic characteristics to mesothelial cells that line other body cavities, such as the peritoneum [7]. The pleural mesothelial cell is the most common cell in the pleural space and is the primary cell that initiates responses to noxious stimuli [6]. The cells produce a number of different molecules, some of which are: Hyaluronic acid, glycoproteins, nitrous oxide and growth factor beta 1 [8]. Figure 1a shows a rudimentary schematic of the lungs, the pleura and the pleural space. In normal circumstances, the pleural membranes are separated by a narrow lining of fluid, which is being produced and reabsorbed by the pleura. The production depends on hydrostatic and oncotic differences in the systemic pressure of the pulmonary circulation and pleural space. Figure 1b presents the balance of forces that regulate the amount of fluid formation in the pleural space. The absorption of fluid from the pleural space is regulated by pleural vessels. In normal physiological conditions, there is 0.26 mL of fluid per kg of body mass in the pleural space [8]. 

Pleural effusion develops, due to increased fluid production, decreased resorption of fluid, or a combination of both. Pleural effusion occurs in more than 1.5 million patients in the United States. The most common reasons for pleural effusion are heart failure, pneumonia and cancer. A clinically important effusion will be visible only after the fluid has exceeded the absorptive capabilities of the lymph vessels [8]. There are many theories discussing the importance of the pleural space. Some of its roles are: Decreased friction between the parietal and visceral pleura during respiration, enabling changes of the lungs during respiration, as well as an enduring negative pressure which prevents the lungs from collapsing [6,8]. Additionally, Ingelfinger, Feller-Kopmann and Light report that the former adaptive mechanism is unique to humans, since most animals do not possess it [8,9]. 

### 2.2. Pathophysiology of Pleural Effusion

Excess fluid formation in the pleural space occurs because of many different reasons, which can be either malignant or benign [5]. When treating pleural effusion, the physician must determine whether it is a transudate or exudate. For further diagnostic purposes, Light criteria, as well as certain additional criteria, may be used (Table 1) [9].

An exudate is defined by a ratio of protein (pleural space)/serum greater than 0.5, when the lactate dehydrogenase (LDH) ratio in the pleural space/serum is greater than 0.6 and when the concentration of LDH in the pleural space/serum is greater than 67% (2/3) of the upper serum limit [8]. Some studies show that the measurement of pleural cholesterol has a higher specificity and sensitivity in comparison with Light criteria [12,13].

Light criteria correctly identify all exudates, whereas transudates are approximately 25% of cases falsely defined as exudates. These mistakes happen mostly in patients with heart failure or liver cirrhosis, who are receiving diuretics [5,8]. In such cases, serum-pleural effusion albumin gradient (SAAG) values can be of additional help [13,14]. A transudate develops because of systemic factors that affect the formation and absorption of fluid. The most common cause being heart failure. The treatment of transudate effusion is aimed at the elimination of the causing agents [5]. The treatment in exudates is, in comparison, much more aggressive. The most common reason being a parapneumonic effusion that develops in pneumonia and MPE the second most common [8]. Malignant diseases cause besides MPE also paramalignant pleural effusion, which is categorized by the absence of malignant cells. The distinction between those two is important because they greatly differ in prognosis and treatment [15]. A high amount of protein and LDH, as well as pH values below 7.32, are characteristic for MPE [11]. The most common reasons for pleural effusion formation are shown in Figure 2.

### 2.3. Pathophysiology of MPE

In normal circumstances, the influx of fluids into the pleural space is balanced with its resorption. This balance must be disturbed in order to produce a pleural effusion. There must be an increase in entry rate and/or a reduction in exit rate. It is likely that both mechanisms contribute to effusion formation. Namely, an isolated increase in entry rate, unless large and sustained, is unlikely to cause a clinically significant effusion because the absorbing pleural lymphatics have a large reserve capacity to deal with excess pleural liquid. Moreover, an isolated decrease in exit rate is also unlikely to cause a large effusion because the normal entry rate is low [16]. The reasons for decreased resorption can be divided into extrinsic and intrinsic factors. Intrinsic factors can interfere with or inhibit the ability of lymphatics to contract (e.g., cancer infiltration into lymphatics, hormonal disbalance, anatomical abnormalities, etc.). When the function of lymphatics is disrupted, but not due to direct damage of the vessels, we speak of extrinsic factors (e.g., limitation of respiratory motion, mechanical compression of lymphatics, blockage of lymphatic stomata, etc.) [16]. 

The same mechanisms apply to MPE formation. Tumor cells infiltrate the pleural space via the hematogenous, direct or lymphatic spread. Accumulation of fluid in the pleural space can be a consequence of tumor growth blocking the lymphatic drainage [1]. Only 55–60% of patients with pleural metastases or lymphatic metastases develop MPE [17]. Why certain patients with pleural metastases do not develop MPE remains a question. However, it is well-known that “wet” pleural carcinosis in comparison to “dry” pleural carcinosis has a worse prognosis and treatment options [18].

It must be stressed that the impact of tumor-host cell interactions became evident with the development of molecular medicine. Researchers proposed that the hyperproduction of pleural fluid from hyperpermeable vessels presents a significant mechanism of MPE formation. An array of different cells and molecules are involved in this process. The effects of these can be divided into three categories: The first group of molecules stimulate the pleural inflammation (e.g., interleukin 2—IL2; tumor necrosis factor—TNF and interferon—INF); the second group of molecules stimulate tumor angiogenesis (e.g., angiopoietin 1 (ANG-1), angiopoietin 2 (AGN–2); the third group of molecules affect vascular hyperpermeability (e.g., vascular endothelial growth factor—VEGF, matrix metalloproteinases—MMP, chemokine (c-c motif) ligand 2—CCL, osteopontin—OPN, etc.) [1,19]. Additionally, studies show that mastocytes have a significant effect on MPE formation. The release of tryptase alpha/beta 1 and interleukin-1β increases the permeability of the pulmonary vessels and induces the activation of the NF-κB transcription factor, which promotes the accumulation of fluid and tumor growth [1,20].

In the last decade, researchers used genome analysis of tumor cells and discovered that tumors which have activating mutations *EGFR*, *KRAS*, *PIK3CA*, *BRAF*, *MET*, *EML4/ALK* and *RET* are connected to increased MPE formation [18,19,21]. *KRAS* mutations are common for distant metastases and *EGFR* mutations for tumors which metastasize via direct infiltration [18]. Mutations in the primary tumor differ from metastases in MPE [22]. This research goes hand in hand with the area of targeted treatment [23].

## 3. Cancer and Malignant Pleural Effusion

Malignant pleural effusion is almost exclusively (95%) caused by metastases in the pleural space. Two-thirds (70–77%) are histologically classified as adenocarcinoma [21]. The effusion presents itself as the first sign of a disease in two-thirds of all cases. Half of these are caused by LC. Patients with hematological cancer and ovarian cancer (OC) in which MPE presents as the first symptom tend to have a better prognosis in comparison to those who develop MPE at a later stage. The longest interval between cancer diagnosis and MPE formation has been reported in BC. However, no matter the time of formation, MPE is universally a bad prognostic sign [24]. A more in-depth overview of MPE incidence in malignant disease can be seen in Table 2.

### 3.1. Lung Cancer

LC is the most common malignant disease in the world. It accounted for 11.6% (2.1 million) of all newly discovered cancer cases in 2018 and is responsible for 18.4% of all deaths caused by cancer. It occurs more commonly in men than in women, and is the most common by incidence, whereas in women, it is ranked third [26]. 

The LC classification is based on the histopathological subtypes. LC is most commonly divided into small cell lung cancer (SCLC) and non-small cell lung cancer (NSCLC). NSCLC accounts for 85% percent of all LC cases and is further divided into adenocarcinoma (25–30%), squamous cell carcinoma (40%), and large cell carcinoma (5–10%) [27]. It has to be pointed out that despite certain similarities in histological appearance, LC subtypes greatly differ in molecular characteristics [28]. These changes happen via DNA alteration, DNA methylation, mRNA expression, microRNA expression, and protein expression mechanisms [29]. The most common biological markers are *EGFR* mutations and *Alk* translocations [28]. 

Squamous cell carcinoma (SCC) of the lung is the most common cause for MPE in men. In 8–15% of all patients with LC, MPE is discovered at the beginning and in 40–50% it is discovered during disease progression. MPE in LC presents ipsilaterally in 90% of cases and bilaterally or contralaterally in 10% [30,31,32]. In SCC tumor cells invade directly into the pleura [31]. The 5-year OS of these patients with MPE is 3% [33]. Therapy selection is based on *EGFR* status [23,31]. The mutation status is determined from samples of the primary tumor because it has been proven that tumor cells that are found in metastases, including in MPE, have different molecular characteristics from the primary tissue [22]. MPE in SCLC presents in 10–38%, and forms due to indirect infiltration of the lymph vessels [34]. 

### 3.2. Breast Cancer

BC is the second most common cancer worldwide, ranked first in incidence in women living either in developed (794,000/year) or developing countries (883,000/year). It accounted for 11.6% of all newly discovered cancer cases in 2018 and is responsible for 6.6% of all deaths caused by cancer (the fifth most common cause). The incidence rate of isolated BC in the world is increasing, due to the implementation of successful preventive public health mechanisms. The result is a lower incidence of disseminated/progressed disease and lower death rates [26,35,36]. There are many types of BC which we can differentiate based on histopathology and molecular, i.e., intrinsic properties. BC is divided into invasive and non-invasive BC. 75% of invasive BC are ductal, 5–10% lobular, 5–7% medullar, 3–5% mucinous, 1–4% tubular carcinoma and some rarer subtypes [37]. The molecular categorization is based on the hormonal receptor status (estrogen receptor-ER and progesterone receptor-PR) and receptor status for human epidermal growth factor (HER2). According to this, we can divide BC into four types: Luminal A (positive PR and ER), luminal B (positive PR, ER and HER2), triple-negative (PR-, ER- and HER2-negative), as well as HER2 positive (PR- and ER-negative, positive HER2) [38]. BC most commonly metastasizes into the bone, liver, brain and lungs [39]. 

MPE occurs in 2–11% of patients with BC, most commonly one-sided, ipsilateral and is a bad prognostic sign. It can manifest years after the initial diagnosis. BC disseminates into the pleural space via the lymph vessels. The OS of patients with BC and MPE ranges from 5 to 13 months [24,40]. MPE is most common in triple-negative breast cancer (TNBC). TNBC quickly progresses, frequently metastases and is the most aggressive of all BC subtypes. Moreover, hormonal, as well as targeted therapy, is not standard treatment options. Metastases occur most frequently between the second and third year after initial diagnosis. Luminal A and B BC subtypes are in comparison with TNBC less commonly associated with MPE [41]. The metastases frequently undergo subsequent mutations and molecular changes. Therefore, for the most appropriate treatment, additional bimolecular assessment is required [38,42]. For prognostic purposes, Ki-67 is evaluated in MPE. Its presence predicts a worse outcome. Elevated levels are seen in 63% of MPE [43].

### 3.3. Ovarian Cancer

OC accounts for 2.5% of all cancer cases in women, but at the same time, for 5% of female cancer deaths because of low survival rates [44]. In accordance with the WHO classification, we distinguish 13 different types of OC; however, 90–95% of them are carcinomas [45]. OC can be, based on histopathological and molecular analysis, divided into five main groups. High-grade serous carcinoma accounts for 70%, endometrioid carcinoma (10%), clear cell carcinoma (10%), mucinous carcinoma (3%) and low-grade serous carcinoma (5%). Malignant epithelial ovarian tumors are classified in accordance with the International Federation of Gynecology and Obstetrics (FIGO) guidelines. Pleural effusion with positive cytology is regarded as stage FIGO IVA. The 5-year OS for patients with localized disease is 92%, whereas patients with a FIGO IV staged disease have an OS of less than 20% [46].

MPE is with 33–53% the most common peritoneal manifestation of epithelial OC [46]. Ovarian tumor cells infiltrate into the pleural space directly via the diaphragm, pleuroperitoneal or hematogenous [47]. Patients with MPE that forms, due to OC have in comparison with patients suffering from a different cancer type the longest survival rate, which is 21 months after the initial diagnosis [48]. 70% of all OC is recognized at a late stage (FIGO III or IV) [49]. In 15% of newly diagnosed patients, MPE is the first clinical sign of disease [47]. Therefore, clinicians must pay attention to typical symptoms of MPE in all OC patients. MPE in OC presents in 77% of cases ipsilaterally, in 23% bilaterally. Common for OC in comparison with other cancers are elevated values of CA-125 and CA-15-3 markers [47].

### 3.4. Lymphoma

Lymphomas are a heterogeneous group of blood cancers characterized by an uncontrolled proliferation of cells of the lymphatic tissue. Lymphomas are categorized into two main categories, Hodgkin’s lymphomas (HL) and the non-Hodgkin lymphomas (NHL) [50]. In 2018 there were 509,590 new cases of NHL worldwide (2.8% of all newly diagnosed cancer cases) and 79,990 new cases of HL (0.4% of all cancers). The incidence of HL is decreasing, whereas the incidence of NHL is increasing. In 2018 248,727 (2.6% of all cancer deaths) patients died, due to NHL and 26,167 (0.3%) due to HL. 

NHL is ranked ninth by cancer incidence in men and tenth in women [26]. MPE occurs in 16–20% of patients with NHL. The effusion frequently forms on the left side. MPE, due to NHL, is most commonly caused by diffuse giant-cell B lymphoma (60%) and in 20% by follicular lymphoma. The proposed pathophysiological mechanisms of action are: (a) Direct infiltration of the pleural space; (b) lymphatic obstruction with infiltration of pulmonal and mediastinal lymph nodes and; (c) obstruction of the *ductus thoracicus*, which leads to the formation of a chylothorax. New studies show that the main mechanisms are infiltration of the pleural space and tumor-host cell interactions [51,52,53]. 

HL has a bimodal peak in incidence for 15–34 years and after 50 years [36,50]. MPE presents in patients with HL in 10–30% at the time of initial diagnosis, in 60% of patients it develops during disease progression [51]. Lymphomas are the most common reason for MPE formation in children and can also occur in cases of primary lymphoma of the pleura [36]. Primary lymphoma accounts for 7% of all lymphomas. It occurs in patients with HIV and patients that have pleural empyema, due to active tuberculosis [51,52,54]. The diagnosis of MPE caused by lymphomas is very difficult, mainly because of the scarcity of cells in the effusion [55]. It is a bad prognostic sign, and the OS after the occurrence is 3–6 months. It has also been observed that 1/3 of patients with MPE and lymphoma are chemotherapy-resistant [21,56].

### 3.5. Malignant Mesothelioma of the Pleura

Malignant mesothelioma is a very aggressive cancer originating from mesothelial cells, which is most commonly found in the peritoneal space and lung serosa, sometimes on the pericardium and tunica vaginalis testis [57]. The most important risk factors are connected to the exposure of mineral fibers (e.g., asbestos). The incidence is still increasing because of delayed onset of the illness (30–50 years after exposure), but heavily varies by geographic location based on the proportion of cases attributable to occupational asbestos exposure [58]. It is estimated that there are about 3000 new mesothelioma cases in the United States each year [59]. Histopathological subtypes of malignant mesothelioma are epithelioid, mixed or biphasic and sarcomatoid mesotheliomas. The epithelioid subtype is the most common (60–80%) and has the best prognosis with an OS of 13.1 months. The sarcomatoid subtype, on the other hand, has the worst prognosis with an OS of 4–6 months [60]. MPE occurs in 54–90% of all malignant pleural mesothelioma cases and forms at an early stage [21,61,62]. The effusion is biologically active, protects the tumor cells from chemotherapy and induces tumor growth. Initial diagnostic procedures in patients with suspected malignant pleural mesothelioma include contrast-enhanced CT of the chest to identify pleural abnormalities and extent of disease, thoracentesis of any existing pleural effusion for cytologic examination, and closed pleural biopsy [63]. According to the latest guidelines, cytologic evaluation of pleural fluid can be an initial screening test for mesothelioma, but it is not a sufficiently sensitive diagnostic test [64]. Whenever the definitive histologic diagnosis is needed, biopsies via thoracoscopy or CT guidance offer a better opportunity to reach a definitive diagnosis [64].

## 4. Diagnostic Procedure

Almost all radiological procedures can help us identify pleural effusions [65]. A summary of the recommendations from the latest ATS/STS/STR guidelines for the diagnosis and treatment of patients with MPE can be seen in Table 3 [3].

### 4.1. Chest X-ray

Changes on the chest X-ray can be seen in the presence of 200 mL pleural fluid in the anterior posterior (AP) view and 50 mL in the lateral view. Most patients with MPE report dyspnea and their chest X-ray shows moderate to large pleural effusion (80%), 10% of patients have massive pleural effusion and 10% have pleural effusion whose volume is less than 500 mL [66]. 

### 4.2. Ultrasound of the Chest 

We can identify pleural effusion with the help of ultrasound (US) of the chest. This method is more sensitive than chest X-ray. It can help with the identification of pleural metastasis and assessing the thickness of the pleural lining. We can usually see relatively small hypoechoic lenticular masses that are touching the chest wall, or large masses with a complex echogenicity. Additionally, US is very efficient when assessing lung expansion after invasive procedures and for quick diagnosis of pneumothorax [66]. In the last years, US-guided thoracentesis has helped minimize common complications [3]. 

### 4.3. Computed Tomography (CT)

The current method of choice to diagnose a pleural disease is contrast-enhanced CT scan [1]. This method helps differentiate between benign and malignant pleural disease. Significant signs that point to the presence of malignant disease are pleural thickening and nodular lesions. Porcel et al. evaluated a CT scan scoring system which included: The presence of pleural lesion > 1 cm, metastases in the liver, lung mass or lung nodule > 1 cm, no loculations, pericardial effusion, and a non-enlarged silhouette of the heart [67]. A CT score of ≥7 is said to predict malignancy with 88% sensitivity and 94% specificity [66]. 

### 4.4. PET Imaging

Commonly used during the staging of malignant disease is fluorodeoxyglucose (FDG) PET imaging. It does not have a routine role in differentiating between benign and malignant pleural effusion. The PET scan can be very helpful in targeting certain anatomical areas of the pleura to biopsy. It is important in cases of mixed disease, such as mesothelioma and pleural asbestosis [66]. 

### 4.5. Thoracentesis

This method is used as a diagnostic and therapeutic tool. The procedure has been modified with the addition of US, which is very functional for targeting certain anatomical areas of the pleura and finding an appropriate entry point (e.g., biopsy). Pleural fluid aspiration should be performed with aseptic precautions. No absolute contraindications are found for thoracentesis. It has been reported that 60 mL of the pleural fluid is adequate for diagnosing MPE; however, in cases when the procedure is both therapeutic and diagnostic, more fluid should be aspirated [68]. There isn’t a consensus on how much fluid can be therapeutically safely aspirated. For diagnostic purposes, on the other hand, it has been stated that when both direct smear/cytospin and cell block preparations are used, ≥150 mL of fluid should be aspirated [68]. The authors believe that the decision on the amount of fluid aspiration has to be made by a well-versed clinician and based on the patients’ clinical status, as well as presenting symptoms.

### 4.6. Biopsy

#### 4.6.1. Blind Closed Pleural Biopsy

Blind closed pleural biopsy is the second method of choice for diagnosing MPE. It is recommended when pleural fluid cytology is non-diagnostic in a patient with suspected MPE. We use the Abrams or Cope needle in this procedure [66]. 

#### 4.6.2. Image-Guided Biopsy

CT-guided or ultrasound-guided biopsy can be used to collect pleural tissue for diagnosis. It has a reported sensitivity of 76–88% and specificity of up to 100% for diagnosing MPE [66]. 

### 4.7. Cytology

Cytology is an approved initial test with a mean sensitivity of 60%, which depends on the underlying primary tumor, sample preparation and the experience of the cytologist [1]. Cytology of pleural fluid is the least invasive and rapidly effective method for diagnosing malignancy. The use of cytology with pleural biopsy improves the diagnostic approach as it increases the sensitivity to 73%. In case of an unclear diagnosis following diagnostic thoracentesis, a blind needle pleural biopsy and thoracentesis are performed. MPE can be diagnosed when a cytological examination of the pleural fluid reveals the presence of malignant cells in the pleural cavity [15]. 

### 4.8. Biomarkers

Biomarkers are biological molecules found in blood, other body fluids, or tissues that are a sign of a normal or abnormal process, or of a condition or disease. The elevated values of some biomarkers in the pleural fluid can direct us to the correct diagnosis [69]. According to Porcel et al., the diagnostic pleural fluid biomarkers for malignancy can be classified as soluble-protein based, immunocytochemical and nucleic-acid based [69]. Recently, several nucleic acids in MPE have been characterized—mesothelin, CEA, CA15-3, CA125, CYFRA 21-1, receptors on the surface of immune cells (CD163^+^ on macrophages), extracellular matrix proteins (OPN, fibulin-3), RNA/DNA levels and sequence, etc. Clinical use of these biomarkers is limited, due to current inadequate validation. However, it has to be stressed that the elevation of the mesothelin in the pleural fluid is a useful indicator of malignancy, including in initially cytology negative effusion. Recently, promising biomarkers are emerging, but before the clinical use of these, further research with a greater number of studies is needed [1,70]. 

### 4.9. MT and Video-Assisted Thoracoscopic Surgery (VATS)

MT and VATS allow direct visualization and biopsy of abnormalities of the pleura, such as nodularity, masses and thickening. The use of US before MT serves for better visualization of the pleural space and has already become a part of routine procedure, as it reduces the time of the procedure and the number of complications. MT with a local anesthetic has a low degree of complications and mortality despite the invasiveness of the procedure itself [66]. 

### 4.10. MPE Diagnostic Algorithm

When suspecting MPE US guided thoracentesis is indicated. In the case of a confirmed diagnosis, we proceed with further treatment procedures—otherwise, we continue with additional diagnostic procedures. We can choose between repeated thoracentesis, image-guided pleural biopsy (especially if MT is not available), and medical thoracoscopy with pleural biopsy [66].

## 5. Management of MPE

Most patients develop dyspnea at rest, and only a small proportion remains asymptomatic. The most important goal of treatment is to relieve dyspnea in a minimally invasive manner. It is also important to minimize repeated procedures, provide the patient with a definitive pleural intervention and reduce the number of hospitalizations [3]. The treatment approach depends on the physical status of the patient, the type of tumor itself and the expected OS [66]. Asymptomatic effusions, regardless of size, do not require specific actions [71]. In most patients with MPE treated with thoracentesis, repeated pleural effusions occur during the development of the disease. Therefore, in patients with poor prognosis, complete pleural interventions are indicated. This ensures long-term relief of pleural effusion symptoms. These include pleurodesis with a chemical agent (tetracycline, doxycycline and bleomycin), talc pleurodesis via thoracoscopy (poudrage) or chest tube (talc slurry), mechanical pleurodesis at surgery, pleurectomy and insertion of indwelling pleural catheters (IPC) [2]. Figure 3 shows the currently valid clinical pathway for treating patients with MPE.

### 5.1. Therapeutic Thoracentesis

Therapeutic thoracentesis is the first step in management and is recommended to be performed in most patients with dyspnea. The advantages of US thoracentesis are improved safety, ease of execution, as well as a reduced number of complications [66,71]. Definitive pleural intervention is intended for patients with slow progression of the disease, patients who have a very short survival or poor performance status [71]. 

### 5.2. Pleurodesis

Pleurodesis merges the parietal and visceral pleura, which leads to the obliteration of the pleural space and prevents the accumulation of pleural effusion. The detailed mechanism of this procedure is unknown, but it is suspected that inflammation or fibrosis via activation of the transforming growth factor beta plays a crucial role. Various agents, such as talc, bleomycin, tetracycline, *corynebacterium parvum* and doxycycline, are used for pleurodesis [66]. This is a very painful procedure for some patients and requires correct analgesic management [2]. 

#### 5.2.1. Pleurodesis with Talc

In the case of pleurodesis with talc, talc can be administered during thoracoscopy via an atomizer (talc poudrage) or in a suspension form (talc slurry) via a chest tube [66]. 

#### 5.2.2. Mechanical Pleurodesis

Mechanical pleurodesis achieves the same as chemical pleurodesis. However, it has been reported that it is a more effective and secure method in comparison to chemical pleurodesis. Also, it is reported to be more effective in relieving symptoms and providing a better quality of life for patients, especially those with MPE, due to breast cancer with a pH less than 7.3 [72,73]. 

### 5.3. Tunneled Pleural Catheter (TPC)

TPC is a silicone tube, placed into the pleural cavity and tunneled subcutaneously with a small cuff. This allows the patients to easily drain pleural fluid at home or at in outpatient setting. It is proven to be a safe symptomatic procedure that improves the quality of life [66]. TPC is an alternative to pleurodesis for the treatment of recurrent MPE [74]. The advantages of the inserted TPC include clinically significant improvement in dyspnea, placement in the outpatient setting, and the ability of patient self-care at home. Spontaneous pleurodesis is expected to occur in approximately 50% of patients. Among those, this happens approximately 60 days after insertion of TPC [8]. TPC is successfully used in patients with trapped lungs and, in comparison with pleurodesis, it causes fewer infections [66]. It was found that the definitive procedure of insertion of TPC or pleurodesis compared to repeated thoracentesis is associated with a lower number of subsequent procedures and complications [71].

### 5.4. Recommendations for the Treatment of MPE in Specific Cases

#### 5.4.1. Patients with Suspected or Known MPE

In patients with suspected or diagnosed MPE, the use of US for the management of pleural interventions is recommended [3].

#### 5.4.2. Patients with Suspected or Known MPE Who Are Asymptomatic

In patients with suspected or known MPE who are asymptomatic, the question arises as to whether or not pleural drainage should be performed. The main advantage of using early intervention is reduced risk of developing non-expandable lungs at a later stage. Nevertheless, according to the latest guidelines, it is recommended that therapeutic pleural intervention is not performed in these patients [3]. 

#### 5.4.3. Use of Large-Volume Thoracentesis and Pleural Manometry in Patients with MPE

Large-volume thoracentesis is a method in which more than one liter of pleural fluid is removed during the procedure. Pleural manometry is used to measure pressure in the pleura or the elasticity of the lungs. Using this method, we can determine whether the lungs will expand after drainage. The advantage of large volume thoracentesis is the confirmation that dyspnea is due to effusion. This is not always evident in the case of small volume thoracentesis. When the patients’ status does not improve after thoracentesis, the physician must find the underlying cause (e.g., pericardial effusion, pulmonary embolism, etc.). The latest data show that 60% of patients will need another treatment procedure within nine days after initial drainage. In patients with symptomatic MPE, large volume thoracentesis is recommended, especially if it is not clarified whether the symptoms are associated with the effusion and/or the lungs are expandable [3]. 

#### 5.4.4. Use of TPC or Chemical Pleurodesis as the First Collection in Patients with Symptomatic MPE with Expandable Lungs without Prior Therapy

Prior to TPCs, the first-choice therapy was pleurodesis with talc. However, TPCs have become the first choice in patients with known non-expandable lungs. According to the latest guidelines, it is recommended to use both chemical pleurodesis and TPC as the first choice in the management of dyspnea [3].

#### 5.4.5. Use of Pleurodesis with Talc Via Thoracoscopy (Poudrage) or Chest Tube (Talc Slurry) in Patients with Symptomatic MPE

In these patients, it is recommended to use talc via thoracoscopy or chest tube, as there is no evidence of a difference in efficacy between them [3]. 

#### 5.4.6. Use of TPC or Chemical Pleurodesis in Patients with Symptomatic MPE with Non-Expandable Lungs with Failed Pleurodesis or Localized Elimination

At least 30% of patients with MPE have non-expandable lungs. Pleurodesis in these patients is unsuccessful in 30%. Therefore, in these patients, the TPC insertion method is the first choice. Compared with pleurodesis with talc, TPC has a shorter hospitalization period [3]. 

#### 5.4.7. Treatment of Patients with TCP Infection

Insertion of TPC has become the method of choice in many patients. The incidence of infections when inserting TPC is low. Infections associated with TPC are normally treated without catheter removal. When the infection does not improve, it is recommended to remove the catheter [3]. 

### 5.5. Other Approaches

#### 5.5.1. Pleurectomy

Radical total or subtotal pleurectomy (resection of the parietal and visceral pleura) by decortication (removal of the fibrin pleural cortex) is occasionally used in patients with MPE where chemical pleurodesis is unsuccessful. In patients with malignant mesothelioma, it is rarely used due to the lack of data on the effectiveness of the intervention in comparison to less invasive procedures. Patients should be suitable for surgery and have a longer life expectancy. Subtotal pleurectomy can be performed with a thoracoscopic approach. The subtotal procedure itself is almost always effective in obliterating the pleural space [3]. 

#### 5.5.2. Shunt

A pleuroperitoneal shunt is rarely used in patients with trapped lungs, malignant chylothorax, or after unsuccessful pleurodesis. The reason for the rare use of a shunt are problems characteristic of established communication (blockage, infections, etc.) and the relative aggressiveness of the intervention compared to TPC. The advantage of a pleuroperitoneal shunt may be in the case of malignant chylothorax. The procedure is performed during thoracoscopy. The insertion has also been performed by a radiologist. Evidence of efficacy is currently mixed, and there is a significant number of reported complications. The use of the technique is currently not a part of the standard clinical pathway and recommendations [3].

#### 5.5.3. Intrapleural Application of Fibrinolytic

The use of fibrinolytic (urokinase) compared to placebo for MPE patients has not been shown to be more effective in randomized trials. The use of the technique is currently not part of the standard clinical pathway and recommendations [3]. 

#### 5.5.4. Antitumor Therapy

Antitumor therapy includes chemotherapy, target therapy and immunotherapy [2]. In most patients anti-tumor therapy is not considered more appropriate than standard treatment for the management of symptoms that occur due to MPE; however, due to the progress of cancer immunotherapeutics over the last decade researches strive to devise new methods to treat rather than palliate malignant pleural effusions [75]. For the time being, no international recommendations have been made about the rationale of using anti-tumor therapy against standard palliative MPE treatment procedures. The response of MPE is poor in systemic anti-tumor therapy in most malignant conditions. There are exceptions because of the effective treatment of the primary tumor (e.g., lymphoma, BC, SCLC, germ cell tumors, prostate cancer and OC). 

It has been reported that the vascular endothelial growth factor (VEGF) functions as an important cytokine in the process of formation of MPE. Studies have shown that some MPEs respond to bevacizumab (a recombinant monoclonal antibody against VEGF). Patients with a mutant EGFR in NSCLC respond to EGFR-tyrosine kinase inhibitors. Unfortunately, most patients who initially had mutant EGFR in NSCLC without pleural effusion develop resistance to the prescribed therapy within a year. Later, MPE often develops. In the future, additional research is needed to confirm the efficacy of bevacizumab and EGFR-tyrosine kinase inhibitors [2].

## 6. Prognosis

MPE is often the first sign of cancer. The presence of MPE in advanced cancer is associated with a poor prognosis [48]. In general, observational studies show that mortality in patients with MPE is higher than in those with metastatic cancer without MPE [48,76]. The prognosis in a group of patients with MPE depends on many factors, such as age, performance score, tumor type, tumor stage, comorbidity, pleural fluid composition and response to therapy. 

In the future, it is expected that the incidence/prevalence of MPE will increase as the global number of patients diagnosed with carcinomas grow and the overall survival in a patient with MPE improves. MPE is characteristic of advanced or metastatic malignancies with a poor survival prognosis, ranging from a median of three months to 12 months depending on the underlying patient and tumor factors [2]. 

The tumor subtype has an important effect on survival. Carcinomas of the lungs and gastrointestinal system have the worst survival prognosis, as the overall survival range from 2 to 3 months. On the other hand, patients with malignant mesothelioma and hematological carcinoma have an overall survival approaching one year [8]. 

In the past few years, several prognostics indicators have been studied to predict the survival of patients with MPE. Currently, the most established method for predicting the prognosis of patients with MPE is the LENT score (Table 4). The LENT score (L—LDH level; E—ECOG; N—neutrophil-to-lymphocyte ratio; T—tumor type) is a clinically significant prognostic method, which helps predict survival and guide management for patients. It is the first validated risk stratification system to predict survival in MPE and is better in predicting survival for an individual patient than ECOG performance status alone [25]. 

The LENT score is calculated based on pleural fluid lactate dehydrogenase, Eastern Cooperative Oncology Group performance score (ECOG), serum neutrophil-to-lymphocyte ratio (NLR) and tumor type. Each of the prognostic indicators has a certain numeric value associated with it. Based on the calculated scores, patients are stratified to low (score 0–1), moderate (score 2–4), or high-risk groups (score 5–7). The low risk group has a median survival of 319 days, as compared with a median survival of 130 days in the medium-risk group and 44 days in the high-risk group [8,25]. Another scoring system is the Brims’ decision tree, which proved to be effective (94.5% sensitivity and 76% positive predictive value) in predicting the prognosis in patients with malignant pleural mesothelioma [77].

## 7. Discussion

MPE is a serious pathological entity common in patients with advanced malignancies with poor prognosis and a median survival of afflicted cancer patients ranging from 3 to 12 months [4,66]. The management of cancer patients is and will remain difficult; however, the new guidelines offer a more patient-centered and -friendly approach which can alleviate some of the burden. Both guidelines are based on the analysis of randomized clinical trials, studies and offer these recommendations, based on the best available evidence [2,3]. Of great importance for patient management are the recommendations in favor of using US to guide pleural interventions, not performing pleural interventions in asymptomatic patients with MPE and using either an indwelling pleural catheter (IPC) or chemical pleurodesis in symptomatic patients with MPE and suspected expandable lung. These recommendations lead to fewer hospitalizations and better patient outcome. The authors would like to note that surgical options for MPE in the form of thoracoscopic abrasion may, based on our studies, in certain scenarios, prove to be equivalent to chemical pleurodesis [72,78]. Of great clinical significance is also the validation of two prediction scores (Brims’ score and Lent score). The Lent score is due to its simplicity attractive for both clinical practice and research settings [2,3].

Open for discussion, further research and of great interest remain a few areas. We know that biomarkers can be classified into predictive biomarkers (e.g., HER2 that predicts response to trastuzumab in breast cancer); prognostic biomarkers and; diagnostic biomarker (e.g., stool DNA testing for colorectal-cancer) [79]. The correct prediction enables us to find adequate therapy, and with a timely diagnosis, the patient has the chance for a better outcome. Despite there being a large number of potential biomarkers, there is only a limited number of prognostic and diagnostic tumor biomarkers that are approved of (e.g., mammaglobulin, CK19—BC diagnosis; BRCA1 and 2 gene mutations—OC diagnosis) [79]. Some examples of potential biomarkers regarding MPE and the underlying diseases are described by Porcel et al. He suggested that pleural fluid may provide an adequate sample for analysis of molecular markers to guide patients with NSCLC to appropriate targeted therapies and offers an overview of different biomarkers [69]. Especially of interest are biomarkers in malignant pleural mesothelioma because of its late onset and long asymptomatic course. [80,81]. Numerous biomarkers have been reported to show promise, including osteopontin, fibulin-3, soluble mesothelin-related proteins, high mobility group box 1, micro-RNA’s, etc. [80]. Worth mentioning are the promising results of the KEYNOTE-028 trial (ClinicalTrials.gov identifier: NCT02054806) using single-agent pembrolizumab, a programmed death protein 1 (PD-1) antibody in patients with malignant pleural mesothelioma. This, in turn, rekindled the interest for the use of immunotherapy in mesothelioma and other advanced solid tumors [82,83]. Nevertheless, to date, no international guidelines recommend the use of anti-tumor medical treatment, before standard palliative procedures for MPE management [2]. Moreover, there are a few other specific areas of interest that have not been fully elucidated. There is a lack of studies that compare palliative procedures for MPE with anti-tumor treatment [2]. Due to MPE being more frequently a late stage complication in malignant diseases, the approaches are currently still mostly focused on the management of symptoms. It remains true that for the most effective treatment, clinicians must be able to identify the underlying disease at an early enough stage (e.g., BC public health interventions, screening programs etc.). Another field of interest with little high-quality evidence is the management of trapped lung in MPE. The authors believe that the use of IPC in such a setting is valuable; however, the use of other modalities (e.g., use of intra-pleural fibrinolytic therapy) may also prove useful in certain cases. Furthermore, there still is not a consensus on the volume of MPE drainage. Traditionally, in a therapeutic setting, fluid removal was discontinued when the total amount of fluid removed reached 1000 to 1500 mL, due to fear of re-expansion pulmonary edema and pneumothorax ex vacuo [84]. Ault et al. showed that the common assumptions regarding thoracentesis safety guidelines are commonly inaccurate [85]. The authors are of the opinion that, currently, there is no absolute maximum volume of fluid that can be safely removed during therapeutic thoracentesis. The decisions regarding how much fluid to remove should be left in the domain of experienced clinicians. 

## 8. Conclusions

In conclusion, the management of MPE has advanced drastically, since the last international guidelines were published. However, it still presents a severe medical condition in patients with advanced malignancy. Recent findings in the field of pathophysiological mechanisms of MPE emphasize the role of molecular factors and mutations in the dynamics of the disease and its prognosis. The updated guidelines are expected to help physicians at diagnosing and treating patients with MPE. However, an individual patient approach is still required. Improved treatment of oncology diseases and a better understanding of pathophysiological mechanisms of MPE in the near future should promote better prognosis for patients with MPE.

## Figures and Tables

**Figure 1 medicina-55-00490-f001:**
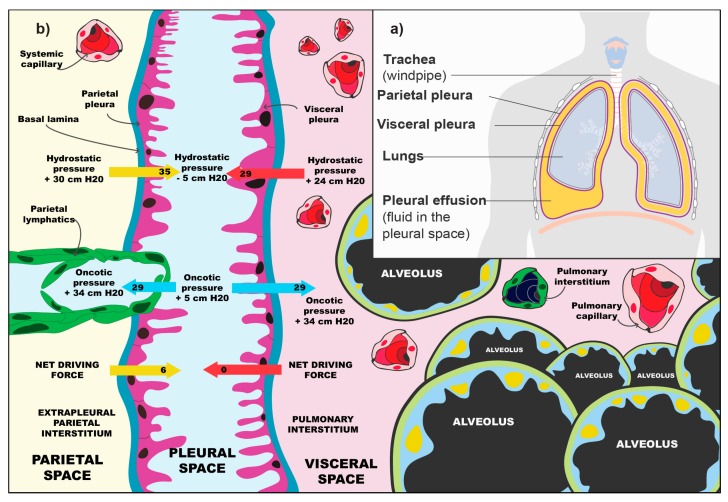
**Schematic of the pleural space in pleural effusion (a) and the physiology of fluid balance (b).** (**a**) Anatomical depiction of the lungs, the parietal and visceral pleura, which surround them and the pleural space with accumulated pleural effusion; (**b**) Visible are the visceral space, the pleural space and the parietal space. The balance of forces depends on the oncotic and hydrostatic pressures. The pleural space has a slightly negative pressure (approx. −5 mmHg), due to the surface tension of the alveolar fluid, the elasticity of the lungs and elasticity of the thoracic wall. This helps in keeping the lungs inflated. Due to higher hydrostatic pressures on the parietal pleura (30 mmHg) than on the visceral pleura (24 mmHg) this leads to fluid production from the parietal pleura. The oncotic pressure is at equilibrium (29 mmHg in both cases). The lymphatic vessels on the parietal pleura are responsible for most of the resorption [8]. The figure (part a) is from the web and marked as »reusable by changing«. (Author: By Cancer Research UK, CC BY-SA 4.0, https://commons.wikimedia.org/w/index.php?curid=34332978). (**b**) adapted from Ingelfinger et al. [8].

**Figure 2 medicina-55-00490-f002:**
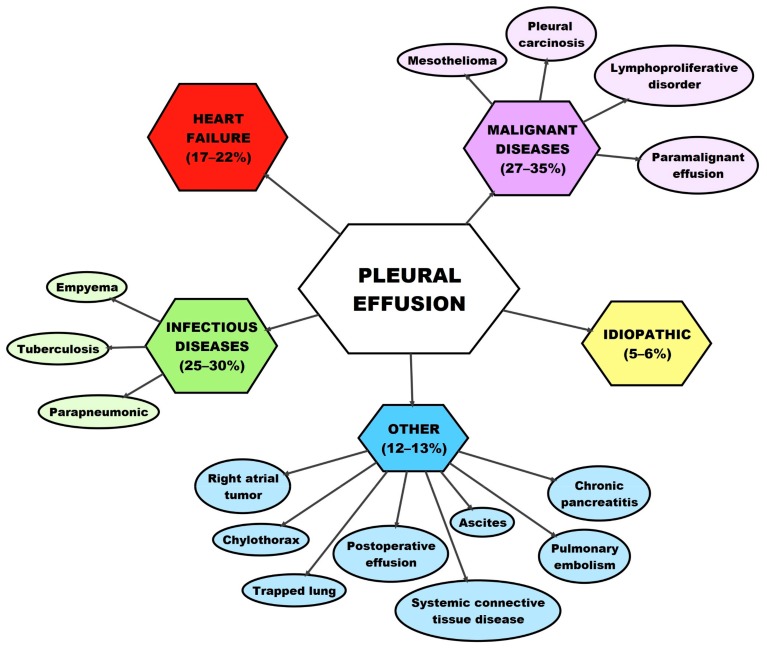
Most common reasons for pleural effusion. Summarized after Nemanič et al. [11].

**Figure 3 medicina-55-00490-f003:**
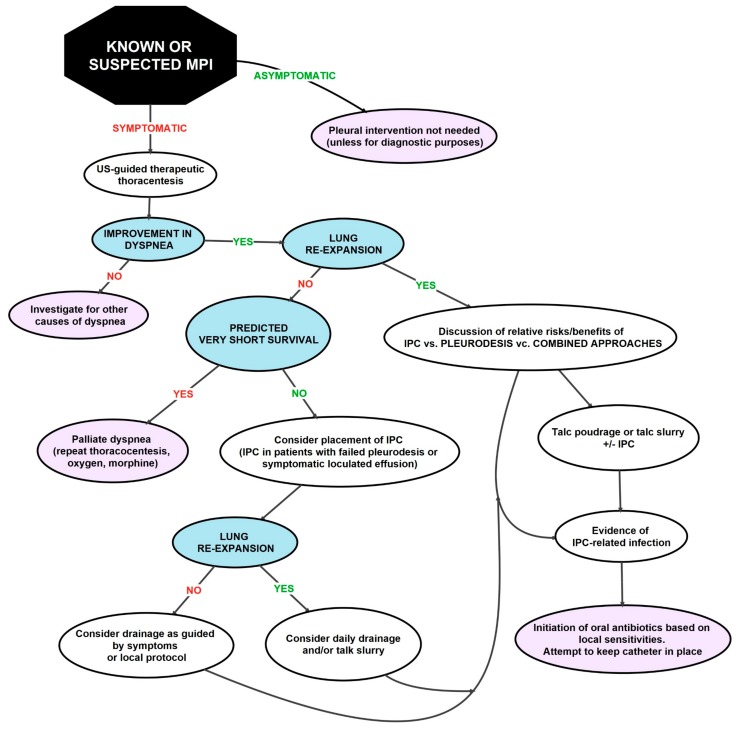
Clinical path for managing MPE. Summarized after Feller-Kopman et al. [3]. Legend: IPC—intrapleural catheter, AB—antibiotic. Colors: Violet—final solution; blue—decision point; black—starting point.

**Table 1 medicina-55-00490-t001:** Criteria for comparison between transudates and exudates.

Criteria	Exudate	Transudate
Standard Light criteria		
PE prot./plasma prot.	>0.5	<0.5
PE LDH/plasma LDH	>0.6 or >2/3	<0.6 or <2/3
Additional criteria		
Gross appearance	Cloudy	Clear
Specific weight	>1.020	<1.020
Protein	>2.9 g/dL	<2.5 g/dL
CHL in pleural fluid	>50 mg/dL	<50 mg/dL
CT radiodensity	4–33 HU	2–15 HU
SAAG	≤1.2 gm/dL	>1.2 gm/dL
Simple overview of specific states		
Empyema	Pus, putrid odor, positive culture.	
Malignancy	Positive cytology.	
Tuberculous pleurisy	Positive AFB stain, culture.	
Esophageal rupture	High salivary isoenzyme form of amylase, low pH (e.g., 6), ingested food fragments.	
Fungal-related effusions	Positive fungal stain, culture.	
Chylothorax	Triglycerides > 110 mg/dL, chylomicrons by lipoprotein electrophoresis.	
Cholesterol effusion	Cholesterol > 200 mg/dL with a cholesterol to triglyceride ratio > 1, cholesterol crystals under polarizing light.	
Hemothorax	Ratio of pleural fluid to blood hematocrit > 0.5.	
Urinothorax	Pleural fluid creatinine to serum ratio always >1, but diagnostic if >1.7.	
Peritoneal dialysis	Protein < 0.5 mg/dL and pleural fluid to serum glucose ratio > 1 in peritoneal dialysis patient.	
Extravascular migration or misplacement of a central venous catheter	Pleural fluid to serum glucose ratio > 1, pleural fluid gross appearance mirrors infusate (e.g., milky white if lipids infused).	
Rheumatoid pleurisy	Cytologic evidence of elongated macrophages and distinctive multinucleated giant cells (tadpole cells) in a background of amorphous debris.	
Glycinothorax	Measurable glycine after bladder irrigation with glycine-containing solutions.	
Cerebrospinal fluid leakage into pleural space	Detection of beta-2 transferrin.	
Parasite-related effusions	Detection of parasites.	

Summarized after Light et al. [9,10,11]. Legend: PE—pleural effusion, prot.—protein, LDH—lactate dehydrogenase, HU—Hounsfield scale, CT—computer tomography, SAAG—serum-pleural effusion albumin gradient, Ht—hematocrit, Glc—glucose, TG—triglycerides, TC—total cholesterol, CR—creatinine.

**Table 2 medicina-55-00490-t002:** Overview of the most common malignant diseases associated with malignant pleural effusion (MPE).

Malignancy	General Median Survival in Days (95% CI)	Histologic Subtype	Prevalence (%)
Lung cancer	74 (60 to 92)	-	-
		Lung adenocarcinoma	29–37
		Small cell carcinoma of the lung	6–9
Breast cancer	192 (133 to 271)	-	-
		Breast adenocarcinoma	8–40
Gynecological malignancy	230 (97 to 279)	-	-
		Ovarian adenocarcinoma	18–20
Gastrointestinal cancer	61 (44 to 73)	-	-
		Gastric adenocarcinoma	2
		Colorectal	1
		Renal cell carcinoma	1
		Pancreatic adenocarcinoma	3
Hematological malignancy	218 (160 to 484)	-	-
		Lymphoma	3–16
Skin cancer	43 (23 to 72)	-	-
		Melanoma	5–6
Mesothelioma	339 (267 to 422)	-	-
		Malignant mesothelioma	1–6
Sarcoma	44 (19 to 76)	Sarcoma	1–3

Summarized after Clive AO et al. [25].

**Table 3 medicina-55-00490-t003:** Summary of current recommendations ATS/STS/STR to treat patients with MPE.

No.	PICO	Recommendations
1	In patients with known or suspected MPE, should thoracic US be used to guide pleural interventions?	Yes.
2	In patients with known or suspected MPE who are asymptomatic, should pleural drainage be performed?	Pleural drainage is not recommended to be performed in this type of patients.
3	Should the management of patients with symptomatic known or suspected MPE be guided by large-volume thoracentesis and pleural manometry?	Yes, large-volume thoracentesis is recommended, as the contribution of thoracentesis prevails over potential complications.
4	In patients with symptomatic MPE with known or suspected expandable lung and no prior definitive therapy, should IPCs or chemical pleurodesis be used as a first-line definitive pleural intervention for management of dyspnea?	Yes, IPC or chemical pleurodesis are used as a first-line definitive pleural intervention for the management of dyspnea.
5	In patients with symptomatic MPE undergoing talc pleurodesis, should talc poudrage or talc slurry be used?	Yes, there was no evidence of differences in efficacy between them.
6	In patients with symptomatic MPE with non-expandable lung, failed pleurodesis, or loculated effusion, should an IPC or chemical pleurodesis be used?	The method of choice is the use of IPC as it is associated with a shorter hospitalization period.
7	In patients with IPC-associated infection (cellulitis, tunnel infection, or pleural infection), should medical therapy alone or medical therapy and catheter removal be used?	Firstly, causative treatment without removing IPC. In case there is no improvement (e.g., persistent infection), the removal of IPC is recommended.

Summarized after Feller-Kopman et al. [3]. US—ultrasound, IPC—indwelling pleural catheter; P—patient, problem or population, I—intervention, C—comparison, control or comparator, O—outcome.

**Table 4 medicina-55-00490-t004:** The LENT score.

Variable			Score
L	LDH level in pleural fluid (IU/L)		
	<1500		0
	>1500		1
E	ECOG PS		
	0		0
	1		1
	2		2
	3 to 4		3
N	NLR		
	<9		0
	>9		1
T	Tumor type		
	Lowest risk tumor types	Mesothelioma Hematological malignancy	0
	Moderate risk tumor types	Breast cancer Gynecological cancer Renal cell carcinoma	1
	Highest risk tumor types	Lung cancer Other tumors types	2
Risk categories			Total score
	Low risk		0 to 1
	Moderate risk		2 to 4
	High risk		5 to 7

Summarized after Clive OA et al. [25].

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
