# Peer review of "Malignant Pleural Effusion and Its Current Management: A Review"

_medicina, 2019, doi:10.3390/medicina55080490_

Round 1
Reviewer 1 Report
Reviewer`s report: received 02.07.2019
Submission Title: Malignant pleural effusion and its current management: a review
Submission ID: 544231
Skok and colleagues have written a comprehensive review covering the different strategies of malignant pleural effusion (MPE) management including its pathophysiology and diagnostic procedures to identify the presence of this effusion .The topics were systematically presented in a well-written manuscript. However, some essential aspects have to be revised to increase the impact of this review as there have been similar publications considering this topic.
Abstract
The authors should explicitly state the objective(s) of the review. This is missing under Background and objectives.
The conclusions mentioned are not the concluding points or remarks based on the reviewed studies/data in the manuscript but rather the objectives of the manuscript.
Fig 1a and 1b. A description of these figures are mandatory. Lines 49-62 do not describe the anatomy and physiology of the illustration shown in Fig 1a.
Fig 1b is quite a busy diagram showing numbered arrows in different directions designated by corresponding hydrostatic pressures, oncotic pressures and net driving forces that regulate the fluid balance. These parameters should be included in the description. Even lines 49-60 are not appropriate to elucidate the events in the figure shown.
3. Cancer and malignant pleural effusion
In my opinion, the statistics for the different types of cancers in Slovenia are not vital on this occasion as this review is not limited to the prevalence of MPE in Slovenia. The data should be limited to a brief description of the cancer its subtypes, and the occurrence or relevance of MPE in each type of tumor.
4.8. Biomarkers
The definition of “biomarkers” is incomplete. Biomarkers are not only found in pleural fluids and are not limited to proteins. Kindly see Clin Pharmacol Ther. 2001 Mar;69(3):89-95 and Mol Oncol. 2012 Apr;6(2):140-6 for references.
7. Discussion
The discussion in its present form is only a summary of selected findings including the biology of MPE and guidelines, which have been cited in the manuscript - some of which have been mentioned three times such as lines 556-558. These have been mentioned in lines 505-508 and 495-499 and are, therefore, redundant.
A clear integrative perspective is lacking. I highly suggest to highlight the major clinical impact of the reviewed studies particularly in the management of MPE. How can these address the current treatment modalities of MPE in LC, BC, OC and malignant pleural mesothelioma? Have there been enough pre-clinical and/or clinical studies to provide an efficient treatment option(s) for MPE? The authors` personal recommendations relative to the current approach in MPE management should be included. This discussion may be included under a subtitle “Clinical Impact” or as appropriate.
Author Response
Dear reviewer,
Please see the attachment.
Thank you.
With best regards,
Kristijan Skok

Reviewer 2 Report
The review gives a practical overview with practice recommendations that are based on pu listed ATS/STS/STR guidelines.
A few minor points- line 112 bold MPE
in line 153 it is stated that most malignant effusion are caused by adenocarcinoma. However line 266 -276 then states that mesothelioma is the most common cause of MPE formation. That is not correct, since lung cancers are so much more common. Also, the correct subtype for mesothelioma is sarcomatous not sarcomatous as per WHO 2015, Finally, diagnosis of mesothelioma by cytology is possible in conjunction with ancillary studies (e.g. BAP1 IHC) AND clinical/radiological correlation (demonstration of invasive tumour by imaging)
In line 310-314, it should be stated how much fluid the authors think is SAFE to aspirate, and may be it could be states that given that sometimes 60 ml are not enough for diagnosis ALL fluid should preferably be submitted for cytological assessment.
Author Response

(The authors gave the same response as above.)

Round 2
Reviewer 1 Report
Submission Title: Malignant pleural effusion and its current management: a review
Submission ID: Revised medicina 544231
Skok and colleagues have appropriately addressed all of the issues raised by the reviewer.
There is one minor concern, which should be addressed:
3.5 Mesothelioma
Based on lines 274-288, the tumor being referred to is malignant pleural mesothelioma. For the sake of clarity the subtitle should be changed to “Malignant pleural mesothelioma”.
Author Response

(The authors gave the same response as above.)
